# DISTANCE-BASED COMPOSABLE REPRESENTATIONS WITH NEURAL NETWORKS

## ABSTRACT

We introduce a new deep learning technique that builds individual and class representations based on distance estimates to contextual dimensions for different modalities. Recent works have demonstrated advantages to creating representations from probability distributions over their contexts rather than single points in a low-dimensional Euclidean vector space. These methods, however, rely on preexisting features and are limited to textual information. In this work, we obtain generic template representations that are vectors containing the average distance between the distribution of a class and that of contextual information, the latter of which consists of a selection of a thresholded number of classes from the power set of classes. These representations have the benefit of being both interpretable and composable. They are initially learned by estimating the Wasserstein distance for different data subsets with deep neural networks. Individual samples or instances can then be compared to the generic class representations, which we call templates, to determine their similarity and thus class membership. We show that this technique, which we call WDVec, delivers good results for multi-label image classification. Additionally, we illustrate the benefit of templates and their composability by performing retrieval with complex queries where we modify the information content in the representations. Our method can be used in conjunction with any existing neural network and create theoretically infinitely large feature maps.

## 1 INTRODUCTION

We introduce a new deep learning technique to create interpretable and composable representations both for generic classes as well as for individual samples based on distance estimates with respect to contextual information. The generic class representations, which we refer to as 'templates', express how samples from a class relate to other classes on average and can be used to efficiently determine class membership.

Most neural network-based approaches to representation learning, focus on learning locations of entities in a low dimensional Euclidean vector space. Word2Vec, for example, extracts meaning from the learned location of words in a vector space (Mikolov et al., 2013a;b). Other text-based approaches use the hidden state vector of LSTM networks (Hochreiter & Schmidhuber, 1997) that learn useful information while performing sequence-to-sequence learning tasks (Sutskever et al., 2014), such as machine translation (Wu et al., 2016), or surrounding sentence reconstruction (Kiros et al., 2015). For images, relevant features are typically extracted from convolutional neural networks (CNNs) (LeCun et al., 1998). When they are pre-trained on a classification task, they are considered to contain abstract knowledge that can be used as an input to perform further tasks (Vinyals et al., 2015; Karpathy & Fei-Fei, 2015; Xu et al., 2015).

Such point-based representations have achieved great results for many tasks, but lack flexibility. Word2Vec representations don't change depending on the context. CNN-based representations fail to accurately relay all relevant features for interacting objects in scenes. Recent works in Natural Language Processing (NLP) have attempted to make point estimations dependent on the context, for example ELMo (Peters et al., 2018) and BERT (Devlin et al., 2018), yet such representations still lack interpretability. Other approaches have started looking at representations that are essentially probability distributions that are built from the co-occurrence of particular contexts. Singh et al.

(2019) create sentence embeddings by defining a Context Mover's Distance over words occurring in different contexts and Wu et al. (2018) create text document embeddings with feature maps by estimating the distance of a document to a range of arbitrarily created documents. In this work we

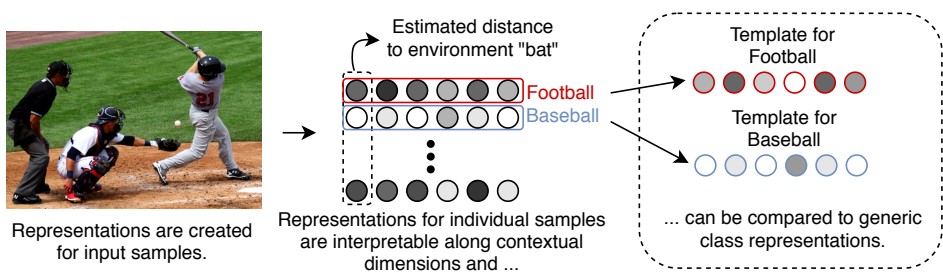

Figure 1: Our method creates representations for individual samples that can be interpreted along contextual dimensions, which we call environments. Those subvectors can then be compared to generic class representations, which we refer to as 'templates'.

will build on the latter approaches by creating a method that is not restricted by modality type. Our approach relies on neural networks to estimate Wasserstein distances between the sample spaces for different *classes* and the sample spaces for events that we call *environments*. These environments are defined by the occurrence of any element in a subset of attributes where each subset, limited to a certain size, is selected uniformly from the power set of all attributes. While in this work we limit the definition of attributes to simply refer to 'class labels', future work might build on a broader interpretation. In the simplest case, when the size of the subsets is limited to 1, each environment is given by the occurrence of a singular class label. The goal is to capture some common sense knowledge about how classes relate to a range of differentiating contexts. Such environments are bound to have distinctive features in relation to any given class as they are made up of a combination of arbitrarily chosen factors. This mechanism loosely resembles the associative nature of human memory. Long-term memory storage is believed to rely on semantic encoding that performs better if it can be associated with existing contextual knowledge (Lepage et al., 2000; Murdock, 1982). Additionally, relating classes to arbitrarily chosen environments increases the probability of encountering distinctive diagnostic features, which should help retrieval, as might be the case with human long-term memory (Nairne, 2002).

We also propose a method to generate *template* representations of classes such that individual input samples can be interpreted in reference to the template representations. A template representation is thus a vector containing the average distance estimates between the sample distribution of a particular class and those of the available environments. This approach, illustrated in figure 1, addresses the limitation of Euclidean point based representations where no distinction is made between generic class representations and specific instance representations. With the use of such templates, class membership of individual samples can then easily be determined by computing the similarity between the representation of an individual sample and all template representations. We will show that, for the same neural network, this method performs well on multi-label classification tasks for images, where any image can be assigned several classes (for example, 'baseball' and 'bat').

Both the template and individual sample representations are interpretable in the sense that they contain distance estimates along contextual dimensions. Samples that belong to a particular class can be analyzed as to how and to what extent they differ from the generic template representation. A sample that is classified as a 'cat' might have a smaller distance to the 'bear' class, suggesting that that particular cat looks more like a bear than the average cat. Another advantage is that we can alter or combine representations in order to modify the information content. The usefulness of these characteristics becomes apparent in retrieval applications. We will demonstrate that it allows one to retrieve an image similar to a particular image but with altered information content.

CONTRIBUTIONS

- We introduce a technique to build representations based on probability distribution estimates that can be used in downstream tasks. The method, which we call WDVec (Wasserstein Distance Vector), is general in that it can be combined with any neural network.

- The obtained representations are interpretable as distance estimates between class and contextual information. They are composable in the sense that they can be modified to reflect different class membership or contextual information. Similar inputs lay near each other in the representation space.

- We define template representations for concepts to which individual samples can be compared. This allows efficient class-membership computation and easy manipulation in the representation space.

## 2    BACKGROUND

We will shortly explain some useful concepts, mostly in relation to distance estimates and how they can be found with neural networks. Such distance estimates will form the basis for our representation learning method further on. We will also refer to recent work that uses similar tactics.

**Useful concepts**    Central to our approach is the concept of the 'Earth Mover's Distance' (EMD) (Rubner et al., 2000), also known as the Wasserstein distance. It can be understood as the minimal amount of effort that is required to move the mass from one probability distribution to another. Finding the EMD between two distributions is traditionally done by solving the 'optimal transport problem' (Hitchcock, 1941; Altschuler et al., 2017; Singh et al., 2019).

While some approaches Kusner et al. (2015); Wu et al. (2018) use some variation of Sinkhorn iterations (Sinkhorn, 1964; Altschuler et al., 2017) to solve the transport problem, neural network-based approximations of the EMD have also been developed in recent years. This is usually in connection to Generative Adversarial Networks (GANs) (Goodfellow et al., 2014) where two neural networks, a generator and a discriminator, are staged against each other. Most recent GAN formulations make use of an Integral Probability Metric (IPM) that essentially defines the loss function for a neural network (referred to as the critic $f$ instead of the discriminator) that allows it to maximally discriminate between two distributions. Most famously, in the Wasserstein GAN (WGAN) formulation (Arjovsky et al., 2017), the IPM of the critic attempts to estimate the Wasserstein distance between the distribution of real samples versus the distribution of generated samples. As the generator improves, the distance estimated by the critic should approach zero. The loss function for the critic of the Wasserstein GAN can be defined as follows:

$$d_{\mathcal{F}}(\mathbb{R}, \mathbb{G}) = \sup_{f \in \mathcal{F}} \left\{ \mathbb{E}_{x \sim \mathbb{R}} f(x) - \mathbb{E}_{x \sim \mathbb{G}} f(x) \right\}. \tag{1}$$

where $\mathbb{R}$ represents the distribution of real samples in a dataset and $\mathbb{G}$ represents the distribution of fake (or generated) samples. $\mathcal{F}$ is a set of measurable, symmetric and bounded real valued functions. If $\mathcal{F}$ is unbounded, equation 1 will scale without bounds.

In order for this approximation to work, $\mathcal{F}$ thus needs to be bounded. Usually, the critic network is restricted to be a 1-Lipschitz function. Several methods have been proposed to enforce this constraint, such as weight-clipping (Arjovsky et al., 2017), gradient penalties (Gulrajani et al., 2017) or spectral normalization (Miyato et al., 2018). The Fisher GAN formulation takes a different approach and bounds $\mathcal{F}$ by construction, that is, by defining a data dependent constraint on its second order moments (Mroueh & Sercu, 2017). The IPM becomes:

$$dFisher_{\mathcal{F}}(\mathbb{R}, \mathbb{G}) = \sup_{f \in \mathcal{F}} \frac{\mathbb{E}_{x \sim \mathbb{R}}[f(x)] - \mathbb{E}_{x \sim \mathbb{G}}[f(x)]}{\sqrt{1/2 \mathbb{E}_{x \sim \mathbb{R}} f^2(x) + 1/2 \mathbb{E}_{x \sim \mathbb{G}} f^2(x)}} \tag{2}$$

Equation 2 can be interpreted as the search for a critic function $f$ that maximizes the average discrepancy between two distributions $\mathbb{R}$ and $\mathbb{G}$ (thus maximizing inter-class variance) whilst minimizing the second order discrepancy (i.e., limiting intra-class variance) (Mroueh & Sercu, 2017). Upon completing training, the numerator thus gives a good Wasserstein distance estimate when inter-class

variance is small in comparison to intra-class variance. When this is not the case, the numerator will be large, yet not an exact approximation of the Wasserstein distance. In practice, the Fisher GAN IPM can be estimated with neural network training where the numerator in equation 2 is maximized while the denominator is expressed as a constraint that is enforced with a Lagrange multiplier.

**Recent work**    The EMD has been successfully applied to NLP problems. Kusner et al. (2015), for example, define the 'Word Mover's Distance' (WMD), which measures the minimal amount of effort to move Word2Vec based word embeddings from one document to another. Their method is interpretable and outperforms other document distance metrics in text-based classification tasks. Wu et al. (2018) improve upon their solution by defining a Word Mover's Embedding, an unsupervised feature representation for documents, created by concatenating Word Mover's Distance estimates to arbitrarily chosen feature maps. They then calculate an approximation of the distance between a pair of documents with the use of a kernel over the feature map. The building blocks of the feature maps are thus documents built from an arbitrary combination of words. This idea is based on the Random Features approximation (Rahimi & Recht, 2008) that suggests that randomized feature maps are useful for approximating shift-invariant kernels. Singh et al. (2019) build on this idea to create unsupervised sentence representations where each entity is a probability distribution based on co-occurrence of words. They embed the distributions in a low-dimensional representation space for text and demonstrate state-of-the-art performance on tasks such as sentence similarity and word entailment. They note that their approach captures uncertainty and allows to interpret the outcome over different contexts. Also relevant is the concept of 'classemes' in computer vision, where rich features are created out of a combination of visual concepts (Torresani et al., 2010). Finally, 'Wasserstein Discriminant Analysis' (Flamary et al., 2018) obtains features through dimensionality reduction by maximizing Wasserstein distances between classes while minimizing intra-class discrepancy.

Our method borrows some concepts from these works, such as the creation of possibly infinite-dimensional feature maps and representations that are built on (randomized) contextual information. WDVec can be used with any modality though and features can be found with neural network-based EMD estimations between different sample spaces. Additionally, we will create generic template representations as well as individual sample representations, which allows efficient membership tests and complex retrieval queries. Our representations will be both interpretable and composable.

## 3    WDVec: Method

We first define some notions that are of use to understand the WDVec method.

### 3.1    Notations

Given a dataset with distribution $\mathbb{P}$, we denote $\mathbb{P}_E$ as the sample space for the event $E$. In this paper, we compare the sample spaces $\mathbb{P}_{c_i}$ of classes $c_i$ with the sample spaces $\mathbb{P}_{e_j}$ of environments $e_j$. Let $n_c$ denote the amount of classes and $n_e$ the amount of environments for the representation spaces that are built in this work. Each environment $e_j$ is composed of the union of $r_j$ attributes $a_k$, where $r_j$ is an integer uniformly chosen from the range $[1, R]$ with $R$ the maximum amount of attributes per environment. In turn, each attribute $a_k$ is uniformly selected without replacement from the set of all possible attributes. As a concrete example, given $R = 5$, one might obtain $r_1 = 2$ and $r_2 = 3$. Environments might then become $e_1 = (a_5 \cup a_{13})$, $e_2 = (a_4 \cup a_9 \cup a_{54})$ and so on. $\mathbb{P}_{e_1}$ would thus refer to the sample space for which attribute $a_5$ or $a_{13}$ occurs. Note that 'attribute' in principle can be interpreted in a general manner, as it can refer to any attribute, class, feature, and so on. We limit the scope in the experiments in section 4, where attributes will simply be classes, i.e., environments will be built from combinations of classes. We leave it to future work to explore other possibilities.

### 3.2    Contextual distance

We propose to represent each sample as a feature map that is constructed with distance estimates between the distributions of classes and environments. Therefore, we calculate the estimated Wasserstein distance $W_{ij}(\mathbb{P}_{c_i}, \mathbb{P}_{e_j})$ between the sample spaces of all classes $c_i$ in $C$ and of all environments $e_j$. Intuitively, one can understand why such co-occurence contains useful information. A subset of image samples containing 'cats', for example, will have a relatively small distance to the subset

containing 'dogs', and a larger distance to the subset containing 'fork'. If one of the environments is defined by the occurrence of 'cats' as well, the estimated distance should be closer to 0. For a concrete example, see table A5 in the appendix. In cases where the estimated distance is not small, the IPM optimization essentially splits samples from the classes and the environments by maximizing the outputs relating to distinctive features of both groups. In the next section we explain how such class representations can be built with neural networks, as well as how representations for individual samples can be built.

### 3.3 IMPLEMENTATION WITH NEURAL NETWORKS

To get estimates of the EMD, we will train a critic $f$ to maximize the Fisher IPM in equation 2. The advantage of the Fisher IPM over other Wasserstein distance estimation methods, is that any neural network can be used as $f$ as long as the last layer is a linear, dense layer. Empirically, the Fisher IPM also leads to more stable and accurate distance estimates. The distance $W_{ij}(\mathbb{P}_{c_i}, \mathbb{P}_{e_j})$ can then be found by maximizing equation 2 over the distributions $\mathbb{P}_{c_i}$ and $\mathbb{P}_{e_j}$. As there are $n_c$ classes and $n_e$ environments, this would require training $n_c*n_e$ critics which is not feasible in practice. Therefore we pass samples through a common neural network $N$ for which the output layer has a dimension of $n_f$. These $n_f$ features are then passed to $n_c*n_e$ single layer neural networks, the outputs of which constitute the estimates for all $W_{ij}(\mathbb{P}_{c_i}, \mathbb{P}_{e_j})$.

During training, any given mini-batch will contain samples for many different $c$ and $e$. For any sample $s$, backpropagation is then performed efficiently by multiplying all $n_c*n_e$ outputs $f_{ij}$ with a mask that is 1 if $s \in \mathbb{P}_{c_i}$ or $s \in \mathbb{P}_{e_j}$, 0 otherwise. The average for a critic, $\mathbb{E}_{x \sim \mathbb{P}_{e_j}}[f_{ij}(x)]$, can be calculated over all samples (*weighted*), or only over the samples where the mask is 1 (*non-weighted*). Depending on the application, the first can improve performance for unbalanced datasets by implicitly including an estimate of the probability of occurrence of a particular class. Algorithm A1 in appendix A.4 explains the algorithm in detail.

For typical GAN training, the loss function receives separate batches of real and fake samples. In our case, to improve efficiency the same batch is used for both $\mathbb{P}_{c_i}$ and $\mathbb{P}_{e_j}$ as the multiplication with the mask guides backpropagation over different contexts. While the last layers impose a slightly larger memory footprint, computing time is barely impacted compared to similar neural network-based methods. Also, for applications where only the compact output representation needs to be stored in a database, such as image retrieval, our method is very efficient. The general approach is illustrated in figure 2. As we will discuss in the next section, to determine class membership, one only needs to perform a similarity calculation between a compact representation and the template representations.

### 3.4 TEMPLATE REPRESENTATIONS

Let's denote $o_{ij} = \mathbb{E}_{x \sim \mathbb{P}_{e_j}}[f_{ij}(x)] - \mathbb{E}_{x \sim \mathbb{P}_{c_i}}[f_{ij}(x)]$. This is effectively an estimate of the distance between the distributions of $c_i$ and $e_j$. This value can be found by saving the average value of this output as all training batches pass through the network during the last training epoch. Thus, for each class $c_i$, the class template representation $T(c_i)$ is then defined as:

$$T(c_i) = [o_{i1}o_{i2}...o_{in_e}] \tag{3}$$

i.e., a vector containing distance estimates between the sample spaces of class $c_i$ and all $n_e$ environments $e_j$. Additionally, the representation for any individual sample $s$ can be defined as the matrix $D(s)$ with $n_c$ rows and $n_e$ columns containing the distance estimates $d_{ij}(s)$:

$$d_{ij}(s) = \mathbb{E}_{x \sim \mathbb{P}_{e_j}}[f_{ij}(x)] - f_{ij}(s) \tag{4}$$

The result is that for an input $s$ with class label $c_i$, $D_{i,:}(s)$ is correlated to $T_{c_i}$ as its distance estimates with respect to all different environments should be similar, whereas this is not the case for a sample without class label $c_i$. Therefore, the cosine similarity between vector $D_{i,:}(s)$ and the template $T(c_i)$ will be large for input samples from class $i$.

Such templates can then be used in a variety of ways, for example in multi-label classification tasks (see section 4.2). Finding the classes to which a sample belongs can simply be calculated

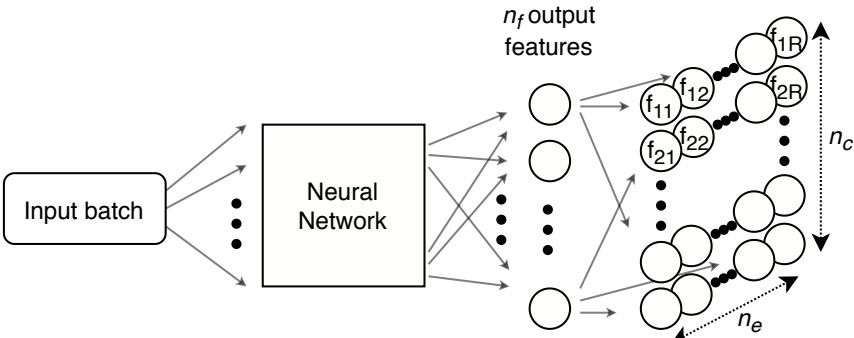

Figure 2: Overview of the WDVec approach. A neural network obtains $n_f$ features from inputs. These features serve as the common input for $n_c * n_e$ critic functions $f_{ij}$. $n_c$ is the amount of classes and $n_e$ is the amount of environments. These critics are trained to estimate the EMD between sample spaces defined by classes $c_i$ and environments $e_j$. $n_c$ template representations $T_{c_i}$ are created by computing $\underset{x \sim \mathbb{P}_{c_i}}{\mathbb{E}} [f_{ij}(x)]$ over all training samples of class $c_i$. Each template has a dimension $n_e$.

by computing the cosine similarity between $D_{i,:}(s)$ and $T(c_i)$ over all classes $c_i$. If they are more similar than a threshold (the level of which is determined during training), the sample is assumed to belong to class $c_i$. Being able to check class membership for all classes is particularly handy for tasks where multiple labels are connected to each sample. Similar to this method, one can also define templates for each column (i.e., environment) dimension and infer which attributes are relevant for the current sample. The template representations thus offer information about how different classes and attributes relate to each other on average. This is shown in table A5.

Another application can be found in image retrieval (see section 4.3). As the representations are interpretable over contextual environments, complex queries can be formulated such as: *Retrieve an image that is similar to a particular sample and that contains a particular class $c_i$ yet doesn't contain a class $c_j$*. Taking this one step further, we will discuss the ability to compose new representations from existing ones in order to modify the content.

## 3.5 COMPOSING NEW REPRESENTATIONS

Given a sample $s \in \mathbb{P}_{c_+}$ and $s \notin \mathbb{P}_{c_-}$ with representation $D(s)$ (see section 3.4). The goal is now to modify $D(s)$ such that it represents a sample $\tilde{s}$ for which $\tilde{s} \notin \mathbb{P}_{c_+}$ and $\tilde{s} \in \mathbb{P}_{c_-}$ while preserving the contextual information of $s$, which in this case reflects how the sample relates to all other classes.

To achieve this, a Singular Value Decomposition of $D(s)$: $D(s) = USV$ is performed, such that the rows of $U$ and the columns of $V$ can be interpreted as the factors leading to the distance estimates as contributed by the corresponding classes $c_i$ and environments $e_j$ respectively. By making relatively straightforward changes to $U$ and $V$ and subsequently reconstructing the information, one is able to compose a new representation $D(\tilde{s})$. To control the extent to which we modify the representation we introduce a factor $0 < q <= 1$, where $q = 1$ is a fully modified representation and $q < 1$ is a weighted combination of the original and modified representation.

Concretely, by increasing the value of $U_{c_+,:}$, one can increase the distance estimate of a sample $s$ with respect to class $c_+$. The opposite can be done for class $c_-$. As we know that $s \in \mathbb{P}_{c_+}$ and $s \notin \mathbb{P}_{c_-}$, it is thus sufficient to simply swap the values of $U_{c_+,:}$ and $U_{c_-,:}$ in order to reverse class membership.

In this study, the attributes (from which the environments are constructed) are simply object classes. In order to change class membership for the sample $s$ we also need to modify all the columns $V_{:,+}$ and $V_{:,-}$ that correspond to environments that are constructed from $c_+$ and $c_-$ respectively. Intuitively, for any sample $s$, the value of the j-th column $V_{:,j}$ that corresponds to an environment $e_j$ is influenced by two factors. The first factor is the amount of classes $r$ in $e_j$. This is easily understood as a larger $r$ means more dissimilar samples and thus a larger distance estimate. The second factor is

whether the current sample $s$ belongs to any of the classes from which $e_j$ is constructed, for example, $e_1 = (c_5 \cup c_7)$ and $s \in \mathbb{P}_{c_5}$. If that is the case, the values of $D_{:,j}(s)$ are mainly determined by the values of $U$ rather than $V_{:,j}$.

A simple strategy is then to set each element of $V_{:,+}$ proportional to the amount of attributes $r_+$ its environment has: $V_{:,+} = (1-q)V_{:,+} + q(V_{:,min} + \chi_+ r_+ (V_{:,max} - V_{:,min})/r_{max})$, where $r_{max}$ is the maximum amount of attributes in any environment. As mentioned before, $q$ is a factor that allows intermediate solutions (thus not fully changing class membership if $q < 1$). Next, we set the value of each element of $V_{:,-}$ to be close to the mean value of each row of $V$ as follows: $V_{:,-} = (1-q)V_{:,-} + q(V_{:,mean} + \chi_- r_- (V_{:,max} - V_{:,min})/r_{max})$. $V_{:,mean}, V_{:,min}$ and $V_{:,max}$ are the mean, minimum and maximum as calculated over the rows of $V$ respectively. $\chi_+$ and $\chi_-$ are independent parameters between 0 and 1 and can be tuned on the validation set. By having $\chi_+ < 1$, one avoids mapping to outliers in $V$. By setting $\chi_-$ relatively small, $V_{:,-}$ will be close to $V_{:,mean}$ and thus have less importance for the reconstruction of the representation than the values of $U$.

A valid representation can then be reconstructed by calculating the outer product $D_{\tilde{s}} = \sum_k \sigma_k U_{:,k} \otimes V_{k,:}^T$ where $\sigma_k$ are the eigenvalues of $D(s)$.

Interestingly, the spectrum of the singular values appears to be non-flat and the eigenvectors belonging to the largest eigenvalue can be visually inspected to determine to which classes the current sample has the smallest distance. In section 4.3 we illustrate this methodology by modifying the information in existing representations and retrieving similar images to the modified representations.

## 4 EXPERIMENTS

To illustrate the approach we perform two types of experiments. First, in section 4.2, we show how it compares to a (binary) cross-entropy baseline for multi-label image classification. In section 4.3, the unique benefits of the representations and their composability are illustrated in a retrieval setting.

### 4.1 SETUP

The experiments are performed on the COCO dataset (Lin et al., 2014) which contains multiple labels for each image. We use all available 91 class labels (which includes 11 supercategories that contain other labels, e.g. 'animal' is the supercategory for 'zebra' and 'cat'). One image can contain more than one class label. We use the 2014 train/val splits where we split the validation set into two equal, arbitrary parts to have a validation and test set for the classification task [1]. Unless noted otherwise, the model is a ResNet-18 with weighted calculation of averages. $n_e$ is 300 and $R$ is 40. $n_f$ is chosen to be 1024. All images are randomly cropped and rescaled to $224 \times 224$ pixels. For all runs, an Adam optimizer was used with learning rate $5.e-3$. $\rho$ for the Fisher GAN loss was set to $1e^{-6}$. In all experiments in this work, the attributes from which the environments are constructed are class labels, e.g. $e_1 = c_{person} \cup c_{car}$. Parameters are found empirically based on performance on the validation set.

### 4.2 CLASSIFICATION

In this experiment, we compare WDVec to an approach that uses cross-entropy for multi-label image classification. The multi-label image classification task is of interest as multiple labels per image need to be identified. The typical approach uses a binary cross-entropy loss to determine whether each label independently should be applied or not. With WDVec, classification is performed by comparing classes to similar contextual environments, that in aggregate contain information about all classes. This is done efficiently through the use of the templates: the cosine similarity is computed between sample representation and template representation and compared to a threshold that is determined during training.

We use some recent state-of-the-art classification models to compare performance: ResNet-18, ResNet-101 (He et al., 2016) and Inception-v3 (Szegedy et al., 2016). For each experiment exactly the same neural networks are used in both approaches where only the last layers are modified.

---

[1] Dataset splits will be published upon acceptance

Table 1: F1 scores, precision (PREC) and recall (REC) for different models for the multi-label classification task. $\sigma$ is the standard deviation of the F1 score over three runs. BXENT refers to binary cross-entropy loss. All results are the average of three runs.

| MODEL | METHOD | F1 | PREC | REC | $\sigma$ |
|---|---|---|---|---|---|
| ResNet-18 | BXENT | 0.517 | **0.677** | 0.418 | $6.3e^{-3}$ |
| ResNet-18 | WDVec | **0.529** | 0.600 | **0.473** | $3.5e^{-3}$ |
| ResNet-101 | BXENT | 0.505 | **0.663** | 0.409 | $1.39e^{-2}$ |
| ResNet-101 | WDVec | **0.538** | 0.595 | **0.494** | $2.6e^{-3}$ |
| Inception-v3 | BXENT | **0.562** | **0.707** | 0.4667 | $9.4e^{-3}$ |
| Inception-v3 | WDVec | 0.554 | 0.550 | **0.559** | $1.0e^{-3}$ |

In table 1, it is shown that WDVec performs better in terms of the F1 score when combined with the ResNet models, and yields similar results with the Inception-v3 model. The performance of WDVec does depend on the choice of the parameters $n_e$ and $R$. Increasing $n_e$, the amount of environments, leads in general to better performance, although it tends to plateau after a certain level. For $R$, the maximum amount of concepts per environment, a value of roughly $n_c/2$ leads to relatively good results. This can be understood in the sense that combining a large amount of attributes creates a unique subset to compare samples with. When $R$ is too large, however, subsets with unique features are no longer created and performance deteriorates. The influence of $n_e$ and $R$ is illustrated in the appendix in figure A1. We also find that, even when $n_e$ and $R$ are small, the outcome is not particularly sensitive with regard to the choice of environments. This suggests that the amount and diversity of environments is more important than the composition of the environments. This is illustrated in the appendix in tables A1 and A2.

## 4.3 RETRIEVAL

This experiment is designed to show the interpretability and composability of the representations. We formulate some retrieval queries that seek to retrieve samples with modified class membership while retaining contextual information as follows: "Given a sample $s$ that belongs to class $c_+$ but not $c_-$, retrieve the sample in the dataset that is most similar to $s$ that belongs to $c_-$ and not $c_+$". $c_+$ and $c_-$ are uniformly chosen in each case from the remaining class labels. For this experiment, we uniformly select 100 samples from the validation set as reference images. Let $cos(x, y)$ be the cosine similarity between two flattened representations $x$ and $y$ and let $mean\_cos(x, y)$ be the mean cosine similarity between $x$ and $y$ with the mean calculated over all class dimensions. We denote the representation for sample $s$ as $D(s)$ as defined in section 3.4. For each reference sample $s_r$ we retrieve from the remaining samples the nearest sample $s$ according to the following methods:

1. **Nearest Neighbor (NN):** $cos(D(s_r), D(s))$
2. **SIM:** $cos(D(s_r), D(s))$ subject to $cos(D_{+,:}(s), T_+) < t_+$ and $cos(D_{-,:}(s), T_-) > t_-$ where $T_+, t_+, T_-$ and $t_-$ are the template representations and thresholds for classes $c_+$ and $c_-$ respectively.
3. **COMP slight:** $mean\_cos(D(\bar{s}_r), D(s))$ over all classes $c$ for which $cos(D_{c,:}(s), T_c) > 0.9 \times t_c$ where $D(\bar{s}_r)$ is a modified version of $D(s_r)$ with a factor $q < 1$ (here: $q = 0.6$).
4. **COMP heavy:** $mean\_cos(D(\bar{s}_r), D(s))$ over all classes $c$ for which $cos(D_{c,:}(s), T_c) > 0.9 \times t_c$ where $D(\bar{s}_r)$ is a modified version of $D(s_r)$ with a factor $q = 1$.

Remark that for methods 3 and 4, the similarity is calculated over class dimensions where classes with low relevance, i.e., those that have a low similarity with the templates, are not taken into account. The templates are thus essential to methods 2,3 and 4. Note also that methods 3 and 4 rely on the methodology of section 3.5 to compose new representations. For these experiments we set $\chi_+$ and $\chi_-$ to 0.2 and 0.07 respectively. The distinction between methods 3 and 4 reflects how the representations have acquired some type of common-sense knowledge: it is not necessarily reasonable to retrieve an image that replaces a train with the category 'orange', as such a request could be interpreted in many ways. Moving to more dissimilar images is thus a reasonable outcome

Table 2: Precision and similarity scores for retrieved images. The baseline consists of CNN features from the last pooling layer of a ResNet-18 architecture. For comparison, the 'NN' method for unaltered queries is added. 'COMP heavy' achieves the highest precision for altered class queries. Our method outperforms CNN features on all accounts in terms of precision. Note: the value of the similarity between CNN and WDVec should not be compared directly as the features differ in size.

|  |  | NN | SIM | COMP slight | COMP heavy |
|---|---|---|---|---|---|
| CNN features | Precision | 0.80 | 0.14 | 0.06 | 0.23 |
|  | Avg sim | 0.85 | 0.78 | 0.74 | 0.70 |
| WDVec | Precision | 0.85 | 0.31 | 0.47 | 0.62 |
|  | Avg sim | 0.97 | 0.81 | 0.64 | 0.60 |

in such cases. As a method to illustrate the sensitivity of the factor $q$, we measure the average $q$ that is needed to modify the class label from the retrieved sample in the COMP method. It turns out that the expected $q$ is 0.505 with a $\sigma$ of 0.026, illustrating both that the composition method works as expected and that class membership isn't sensitive to small perturbations.

As a baseline we retain CNN features of size 512 from the last average pooling layer of the ResNet-18 model. To compare them to WDVec, we define templates for the CNN as the average feature vector for a particular class. The 'SIM' method can be directly applied. For the 'COMP' methods, we modify the features of a sample $s$ by subtracting the template of $c_+$ and adding the one of $c_-$.

The advantages of the composability of the representations become obvious in table 2. The precision is shown, which is determined by whether the retrieved sample belongs to the desired class(es), as well as the average similarity between retrieved image and queried image. The 'SIM' method was often able to find very similar samples that were misclassified however, thus leading to a relatively low precision score. With the 'COMP' methods a better balance between similarity and precision can be found. Some examples of retrieval results are presented in appendix A.3. The obtained representations can thus be interpreted, composed, and capture useful information such that similar instances are near each other in the representation space. The templates are useful to interpret class membership efficiently and manipulate the instance representations as demonstrated here for a retrieval task. We see potential additional uses for the templates in future work, for example as a reference representation that retains knowledge in a continual learning setting. Additionally, further research might explore how the method can be applied to other tasks and modalities with alternative building blocks for the environments.

## 5 CONCLUSION

Our main contributions are firstly the introduction of a technique to build representations that rely on distance estimates that can be combined with any neural network. This is demonstrated by performing multi-label image classification with different state-of-the-art models and achieving good results. Secondly, the representations are interpretable and composable in the sense that class membership and contextual information can be observed and modified by means of a singular value decomposition. This is shown to be useful in a retrieval task where the class membership of samples is modified while contextual information is maintained. Samples that are altered achieve a better trade-off between precision and similarity than unaltered samples. Finally, we introduce the concept of template representations which are generic class representations. We show how they lead to efficient and accurate class membership calculation in a multi-label classification experiment. Additionally, they help achieve good precision in the retrieval task when representations are modified along class dimensions. The distance estimates in the templates also provide an overview of how different classes relate to each other.

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

# A    APPENDIX

## A.1    HYPERPARAMETER SELECTION FOR ENVIRONMENTS

In image A1, the influence of $R$ and $n_e$ on performance is shown for multi-label image classification. In table A1 and A2, the standard deviations of the F1 scores are given for different values of $n_e$ and $R$ respectively. The standard deviation is smaller than or equal to $1\%$ indicating low sensitivity to different choices of environments. Also shown in tables A3 and A4 are the average values of the spectral norms over 100 arbitrarily selected representations, for varying hyperparameters $n_e$ and $R$.

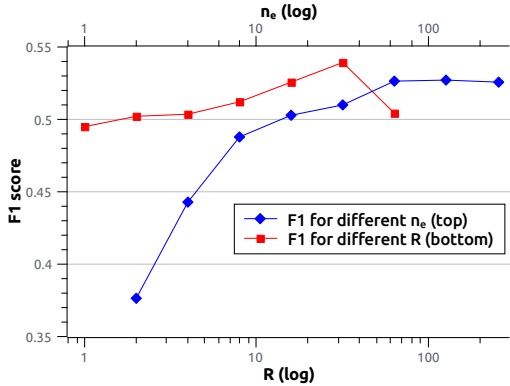

Figure A1: Influence of $R$ and $n_e$ on the F1 score for multi-label image classification using the WDVec approach with ResNet-18. When modifying $R$, $n_e$ is fixed to 300. When modifying $n_e$, $R$ is fixed to 40. All datapoints are the average of three runs.

Table A1: Standard deviations of F1 scores of the classification experiment for different values of $n_e$. Each value is computed on the basis of three runs.

|  | $n_e = 2$ | $n_e = 4$ | $n_e = 8$ | $n_e = 16$ |
|---|---|---|---|---|
| $R = 1$ | $3.6e^{-3}$ | $9.1e^{-3}$ | $9.5e^{-3}$ | $7.5e^{-3}$ |

## A.2 DISTANCES BETWEEN DIFFERENT CLASSES

Table A5 shows some rows and columns of a template representation where $R$ is 1, i.e., every environment consists of 1 particular class. The results show that related concepts have smaller estimated distances than those that are less related. Note that the class 'mouse' only appears in the dataset in relation to computers, rather than animals, which is reflected in the estimates. Some interesting things happen as well, as presumably some of the bananas in the dataset appear in images with computers or similar contexts, thus leading to a smaller estimated distance between the classes 'keyboard' and 'banana'. The distance between the class 'vehicle' and itself is accurately estimated as exactly 0.

## A.3 RETRIEVAL EXAMPLES

In figure A2, some results of retrieval of images are shown. The goal of the 'SIM' and 'COMP' methods is to retrieve an image where the class of the retrieved image reflects the content change, while maintaining the context of the original sample. The original classes in this case were respectively: sheep, zebra, toilet, train. They were modified into the following classes respectively: bear, giraffe, airplane and orange. For the SIM method, similar images are often obtained as only results are returned when the class membership, as determined by the cosine similarity with the templates, allows for it. The method performs badly when the classification fails, as for example in the second row of figure A2. It often leads to very good results though, as in the third row where the 'SIM' method retrieves an image of a bathroom stall from an airplane or in the fourth row where an orange truck is retrieved that is not a train. The 'COMP slight' method reflects an intermediate trade-off between similarity and modified class membership. 'COMP heavy' modifies class membership correctly more often, at the cost of similarity.

## A.4 ALGORITHM

In algorithm A1 we shortly explain the exact algorithm to calculate the backpropagation during the training phase.

Table A2: Standard deviations of F1 scores of the classification experiment for different values of $R$. Each value is computed on the basis of three runs.

|            | $R = 1$ | $R = 2$ | $R = 4$ | $R = 8$ | $R = 16$ |
|------------|---------|---------|---------|---------|----------|
| $n_e = 1$  | $3.6e^{-3}$ | $4.2e^{-3}$ | $5.1e^{-3}$ | $1.3e^{-3}$ | $1.0e^{-2}$ |

Table A3: The average of the spectral norm was taken over 100 arbitrarily selected representations for different values of $n_e$.

|            | $n_e = 4$ | $n_e = 16$ | $n_e = 64$ | $n_e = 256$ |
|------------|-----------|------------|------------|-------------|
| $R = 40$   | $9.1e^3$  | $2.8e^4$   | $1.5e^5$   | $6.7e^5$    |

Table A4: The average of the spectral norm was taken over 100 arbitrarily selected representations for different values of $R$.

|             | $R = 1$ | $R = 4$ | $R = 16$ | $R = 32$ |
|-------------|---------|---------|----------|----------|
| $n_e = 300$ | $2.6e^5$ | $6.6e^5$ | $7.8e^5$ | $5.3e^5$ |

Table A5: Example of a template representation with $R = 1$. Classes that are related have smaller estimated distances. Rows are classes whereas columns are environments that are made up of 1 attribute (in this case an attribute is a class).

|              | VEHICLE | SHEEP | BANANA | TRAIN | MOUSE | SPORTS BALL |
|--------------|---------|-------|--------|-------|-------|-------------|
| PERSON       | 5.09    | 5.11  | 4.99   | 4.60  | 5.06  | 4.32        |
| VEHICLE      | 0.0     | 5.27  | 5.23   | 5.24  | 5.30  | 5.26        |
| ANIMAL       | 5.21    | 3.92  | 5.23   | 5.12  | 5.15  | 5.15        |
| BASEBALL BAT | 5.28    | 5.26  | 5.19   | 5.18  | 5.18  | 3.80        |
| KEYBOARD     | 5.27    | 5.20  | 3.86   | 5.17  | 2.99  | 5.13        |

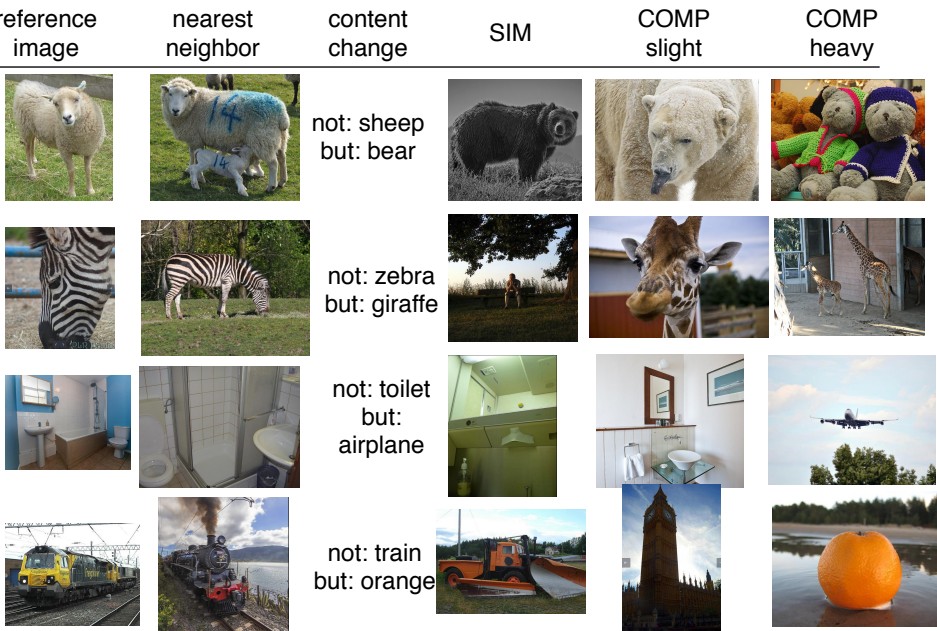

Figure A2: Some results for image retrieval with the different methods. The column 'nearest neighbor' shows the most similar image without modification of the content. The column 'content change' refers to how the class of the representations was altered. The nearest neighbors to the altered representations are then retrieved with the 'SIM', 'COMP slight' and 'COMP heavy' methods respectively.

**Algorithm A1** Algorithm of the training process. Note that in this work the attributes are class labels. For matrices and tensors, $\times$ refers to matrix multiplication and $*$ refers to element-wise multiplication.

For each environment uniformly sample $r$ from the range $[1, R]$ and uniformly select $r$ attributes from the set of all attributes. Create a random feature vector $v$ with shape $[n_c, n_e]$ which has a value of 1 for each uniformly selected attribute. Each column thus has maximum $R$ non-zero entries.

Initialize $\lambda$ as a tensor of zeros with shape $[n_c, n_e]$.

**while** Training **do**
  Sample a mini-batch $b$, with batch size $n_b$, containing samples $s$ and one-hot labels $l$, with shape $[n_b, n_c]$.

  **Create masks**
  Expand $l$ to create a mask $m_c$ with shape $[n_b, n_c, n_e]$, such that $m_{ck,i,:} = 1$ if the $k$-th sample, $s_k$, belongs to class $i$, 0 otherwise.

  Multiply $l$ and $v$, then expand the result, to create a mask $m_r$ with shape $[n_b, n_c, n_e]$, such that $m_{rk,:,j} = a$ where $a$ is the sum of all the attributes of the $k$-th sample that are present in environment $j$.

  **Calculate the FISHER GAN loss**
  Propagate $b$ through the neural network to obtain $out_{logits}$ with shape $[n_c, n_e]$.

  Calculate $out_P = out_{logits} \times m_r$ and $out_Q = out_{logits} \times m_c$.

  $E_{Pf} = sum(out_P, dim = 0)/sum(m_r, dim = 0)$
  $E_{Pfs} = sum(out_P * out_P, dim = 0)/sum(m_r, dim = 0)$
  $E_{Qf} = sum(out_Q, dim = 0)/sum(m_c, dim = 0)$
  $E_{Qfs} = sum(out_Q * out_Q, dim = 0)/sum(m_c, dim = 0)$
  $constraint = 1 - (0.5 * E_{Pfs} + 0.5 * E_{Qfs})$

  Minimize the loss $loss = -torch.sum(E_{Pf} - E_{Qf} + \lambda * constraint - \rho/2 * constraint^2)$
**end while**

