# OpenReview forum: "Distance-based Composable Representations with Neural Networks"
_ICLR.cc/2020/Conference — Reject_

### Official Review · AnonReviewer2 · 2019-10-20
**Official Blind Review #2**

**Rating:** 3

**Review:**

Paper contributions
=================
- This paper proposes a method for constructing representations using a matrix of Wasserstein distances. These distances measure the discrepancy between each class and each environment, that is a random combination of some classes.
- The paper evaluates this approach on a task of image retrieval.

General notes
============
The general idea of measuring the distribution divergence for a set of classes is interesting and seems to be novel. But I argue that this representation can be limiting:
- A set of divergences doesn't contain any pixel-level information, only divergences to some predefined classes
- As a consequence, this representation will not be able to discover information that is not covered by the labels
Because of these limitations, it seems that this particular representation may be less useful for some applications than others.

I don't follow why the paper proposes to use 'environments'  -- random combinations of classes. It seems that a square matrix (n_c x n_c) with all classes should do the same job.

The experimentation is very weak and does very little to support the claims. The paper considers only one substantial task to test the representation. This task is image retrieval by image query. The paper doesn't provide any comparison to existing methods or simple baselines.

The second contribution that the representations are interpretable and composable is not addressed.  I seems that it should be hard to interpret a large vector of distances to randomly chosen subsets of classes. There is no experiment demonstrating interpretability of the proposed approach. The compositionality is not addressed either. The samples provided in the appendix are not convincing.

The paper is generally well written and it is easy to follow. The literature review can be improved by providing prior work where "approaches use hidden state vector of LSTM" and "features extracted from CNNs" instead of generic references.

Some of the claims are vague and excessively broad:
- The proposed technique can be used with any task, but the paper is clearly limited to the retrieval task
- The environments are too vaguely described and can be misinterpreted in the introduction

Conclusion
=========

I recommend to reject on the basis that
- the approach is more limited than the paper advocates
- the experimentation is weak
- some claims are not addressed

Other notes
==========
I recommend using term divergence instead of distance when it is not symmetrical.

**Experience Assessment:**

I have read many papers in this area.

**Review Assessment: Checking Correctness Of Derivations And Theory:**

I carefully checked the derivations and theory.

**Review Assessment: Checking Correctness Of Experiments:**

I carefully checked the experiments.

**Review Assessment: Thoroughness In Paper Reading:**

I read the paper at least twice and used my best judgement in assessing the paper.

---

> ### Author Response · Authors · 2019-11-14
> **Reply to reviewer #2**
>
> “The general idea of measuring the distribution divergence for a set of classes is interesting and seems to be novel.”
> Thank you for your review and the effort you put into it.
>
> “[...]
> - A set of divergences doesn't contain any pixel-level information, only divergences to some predefined classes
> - As a consequence, this representation will not be able to discover information that is not covered by the labels
> Because of these limitations, it seems that this particular representation may be less useful for some applications than others.”
> We would argue that, to the contrary, using labeled information is useful and many applications require such representations. Our approach offers an easy method to integrate continuous (images) and discrete (labels) information into one representation which, for instance, offers many possibilities given the current state of object recognizers in computer vision. In particular, using labeled data to explore both global and local similarities over different classes can be useful in many specialized or critical applications where wrong classification results are highly undesirable (medical diagnosis, robotic automation,...)
>
> “I don't follow why the paper proposes to use 'environments' -- random combinations of classes. It seems that a square matrix (n_c x n_c) with all classes should do the same job.”
> Note that the experiments that are detailed in section 4.2 with plot in appendix A.1 experimentally confirm the validity of using random combinations. Results for $R$=1 are indicative of the outcome when using a square matrix, yet the results improve significantly for larger $R$ values. We also explicitly tried using a square matrix which did not work as well. We have clarified why we use environments in the revised submission (sections 2 and 4.2). In short, we follow the findings of the work of [1] that apply the theory of Random Feature Approximations [2] which has shown to lead to beneficial shift-invariant properties.
>
> “The experimentation is very weak and does very little to support the claims. The paper considers only one substantial task to test the representation. This task is image retrieval by image query. The paper doesn't provide any comparison to existing methods or simple baselines.”
> With all due respect, this is not correct on two accounts: first, we perform not only retrieval experiments (section 4.3), but also classification experiments on the obtained representation (4.2). Additionally, for both experiments we provide a comparison to baselines. For the classification task we provide baselines with three state-of-the-art classification models that employ binary cross-entropy (see table 1, section 4.2), for the retrieval based on modified representations we provide a baseline on the basis of CNN features (see table 2, section 4.3).
>
> “The second contribution that the representations are interpretable and composable is not addressed. I seems that it should be hard to interpret a large vector of distances to randomly chosen subsets of classes. There is no experiment demonstrating interpretability of the proposed approach. The compositionality is not addressed either. [...]”
> We respectfully disagree on this account as well. The retrieval experiment in section 4.3 is specifically designed to illustrate the interpretability and composability following the method described in section 3.5. In that experiment we  compose new representations from existing representations by exploiting the structure of the representations that are interpretable over rows and columns. We subsequently retrieve images that are similar to the modified representations, which we quantitatively have evaluated in table 2 and qualitatively in figure A2. Additionally, the ‘SIM’ retrieval method for this experiment relies on the interpretability of representations over different classes.
>
> “The paper is generally well written and it is easy to follow. The literature review can be improved by [...]”
> Thank you, we have improved this section in the revised version by adding several relevant works.
>
> “- The proposed technique can be used with any task, but the paper is clearly limited to the retrieval task
> - The environments are too vaguely described and can be misinterpreted in the introduction”
> Again we note that we implemented both a classification task and retrieval task. We have rephrased and detailed certain parts of the paper.
>
> “I recommend using term divergence instead of distance when it is not symmetrical.”
> Could you specifically refer to a relevant instance? All distance estimates employed in the paper are intended as approximations or estimates of the Wasserstein distance which is a distance between distributions. The IPM formulation is found to be symmetric and satisfies the triangular inequality when Lipschitz continuous [3].
>
> [1] https://arxiv.org/abs/1811.01713
> [2] http://papers.nips.cc/paper/3182-random-features-for-large-scale-kernel-machines.pdf
> [3] https://arxiv.org/abs/1701.07875

---

### Official Review · AnonReviewer3 · 2019-10-22
**Official Blind Review #3**

**Rating:** 3

**Review:**

The paper defines a representation learning strategy based upon estimation
of a matrix of Wasserstein distances.

The idea is excellent.  The ability to "solve" IPMs reliably is a recent
development in deep learning whose ramifications are still being explored.
Intuitively this line of research could plausibly result in general
methods which are theoretically intelligible and broadly applicable.
Indexing at least one side of the matrix of estimated WDs with events
(rather than classes) has interpretability properties useful for
information retrieval and also conveys benefits reminiscent of learning
with privileged information.

However, the exposition could be greatly improved by using the
standard language of probability theory.  The discussion in 3.1
was particularly painful to read.  What is the difference between
"existing in an environment" and "conditioning on a measurable event"?
Phrases like "belonging to any random subset of the dataset" suggest
a non-deterministic method of selecting an element of the power set of
the training data, but it is unclear what to do if more training data
arrives in this case.

Throughout the entire paper the word "random" is apparently used in the
colloquial sense of "arbitrary".  *Correct every instance of this.*
If you actually are referring to generating samples from a distribution,
be explicit about the generative process.

Section 3.5 was more confusing than enlightening.  In general I understand
that environments can be leveraged for intelligibility and admit manipulation
for information retrieval.  The exact strategy remains somewhat opaque.  If
you are under space constraints refer to an appendix with more explicit details.

In the experiments section phrases like "environments consist of random
combinations of classes" is also not helpful.  Do you mean something like
"uniformly selected from the set of all class pairs?"  Or something like
"uniformly selected from the power set of all classes?"  How volatile
are the experimental results with respect to the non-deterministic choice
of environments?

I want to accept this paper if the exposition is improved, which I think
is possible during the response period.

My other comments are not blocking issues, but would either improve the
current paper or inform future directions of research.

The technique bears some resemblance to Wasserstein Discriminant
Analysis.[1]  That paper seeks a projection that maximizes the ratio of
Wasserstein distance between classes vs. within classes.  Here,
although the common representation is a nonlinear mapping
analogous to a projection, we merely try to estimate all the
Wasserstein distances rather than maximize them, so it is not trained
to be discriminative per se.   That is ok since the representation is
designed to be used for a variety of tasks (modulo section 4.2), but it
does leave open the question "what if the matrix of estimated
Wasserstein distances isn't informative, e.g., due to poor choice of
environments?"  There is no attempt to assess the representation
except via utility in downstream tasks.

The common representation was justified computationally, but I suspect
is beneficial statistically.  It might facilitate safely including a
large number of environments and then spectrally compressing (i.e., SVD)
the resulting matrix without overfitting the data.  However clearly if
the capacity of this layer is too small, then all estimated WDs will
be close to zero.  If we posit a low Bayes error classifier for the
multi-class problem associated with the dataset, that might imply there
is some conditioning of the input under which the matrix of (actual) WDs
has rank equal to the number of classes, which would in turn provide a
useful diagnostic to guard against an insufficiently discriminative choice
of environments or insufficient capacity in the common representation. If
the matrix is full rank with a flat spectrum, however, that might indicate
the choice of environments is too granular and overfitting has occurred,
it's not immediately obvious to me how to guard against this.

I am curious what the results in appendix A.1. look like relative to the spectral
norm or the smallest eigenvalue of the estimated WD matrix (smallest
eigenvalue assuming number of environments < number of classes,
otherwise the k-th eigenvalue where k = number of classes).

[1] https://arxiv.org/abs/1608.08063


**Experience Assessment:**

I have published one or two papers in this area.

**Review Assessment: Checking Correctness Of Derivations And Theory:**

N/A

**Review Assessment: Checking Correctness Of Experiments:**

I did not assess the experiments.

**Review Assessment: Thoroughness In Paper Reading:**

I read the paper at least twice and used my best judgement in assessing the paper.

---

> ### Author Response · Authors · 2019-11-14
> **Reply to reviewer #3**
>
> We thank reviewer #3 for the elaborate and very constructive review. We appreciate that you find it an excellent idea, yet acknowledge that the exposition had some issues. We have tried to address your concerns and have thoroughly improved the formulation in the revised version (especially section 3).
>
> “However, the exposition could be greatly improved by using the standard language of probability theory. The discussion in 3.1 was particularly painful to read. What is the difference between "existing in an environment" and "conditioning on a measurable event"? Phrases like "belonging to any random subset of the dataset" suggest a non-deterministic method of selecting an element of the power set of the training data, but it is unclear what to do if more training data arrives in this case.”
> Indeed, these phrases lacked clarity and we have modified these sentences throughout the whole text in the revised version and especially in section 3.1.
>
> “Throughout the entire paper the word "random" is apparently used in the colloquial sense of "arbitrary". *Correct every instance of this.* If you actually are referring to generating samples from a distribution, be explicit about the generative process.”
> We have corrected our use of the word “random” in the paper. Our intention is to convey the following: Environments are randomly generated in the following manner: the size is uniformly selected from the range given by [1,$R$] and the attributes that make up the environments are uniformly selected without replacement from the set of all attributes.
>
> “Section 3.5 was more confusing than enlightening. In general I understand that environments can be leveraged for intelligibility and admit manipulation for information retrieval. The exact strategy remains somewhat opaque. If you are under space constraints refer to an appendix with more explicit details.”
> We have rephrased and detailed section 3.5 to improve clarity.
>
> “In the experiments section phrases like "environments consist of random combinations of classes" is also not helpful. Do you mean something like "uniformly selected from the set of all class pairs?" Or something like "uniformly selected from the power set of all classes?"
> We improved this formulation (as well as other phrases) as you have suggested. These improvements can be found throughout the text and especially in the introduction and section 3.
>
> “How volatile are the experimental results with respect to the non-deterministic choice of environments?”
> As can be seen from the standard deviations in table 1, the F1 scores in the classification task are not impacted much for different non-deterministic choices. Intuitively, the sensitivity depends on the values of the hyperparameters $n_e$ and $R$ , the amount and maximum size of the environments respectively. We thus added additional sensitivity analyses in appendix A1 (tables A1 and A2) of the revised version, illustrating that the sensitivity remains low for all values of $R$ or $n_e$.
>
> “The technique bears some resemblance to Wasserstein Discriminant Analysis.[1] [...] That is ok since the representation is designed to be used for a variety of tasks (modulo section 4.2), but it does leave open the question "what if the matrix of estimated Wasserstein distances isn't informative, e.g., due to poor choice of environments?" There is no attempt to assess the representation except via utility in downstream tasks.”
> Thank you for this reference, we have added it in section 2 (background – recent work). As mentioned above, we added figures that show a small variance for classification outcomes for different values of $R$ or $n_e$, which suggests the representations are quite robust with respect to the choice of environments.
>
> “The common representation was justified computationally, but I suspect is beneficial statistically. [...]”
> These are interesting points. From our composition experiments we found that, for the given values of $R$ and $n_e$ , the spectrum was very non-flat which suggests indeed that the representation could be further compressed to a large degree.  We believe that your suggestion of a diagnostic to guard against insufficient capacity is very interesting and could be part of further future work, for example by evaluating the evolution of the spectrum of the representations as training progresses.
>
> “I am curious what the results in appendix A.1. look like relative to the spectral norm or the smallest eigenvalue of the estimated WD matrix (smallest eigenvalue assuming number of environments < number of classes, otherwise the k-th eigenvalue where k = number of classes).”
> We have added tables A.3 and A.4 in the appendix that show average values of the spectral norm of 100 representations for different values of $R$ and $n_e$.

---

### Official Review · AnonReviewer1 · 2019-10-23
**Official Blind Review #1**

**Rating:** 6

**Review:**

The authors proposed a template-based interpretable representation that works based on the earth mover's distance of each class to a number of "environments", which could be taken as union of a few random classes. To achieve this, they train several critics based on Fisher GAN. The method is evaluated based on classification and retrieval tasks.
The representation, by construction, is aimed towards interpretation and is specially useful in multi-class classification tasks.
Here are my concerns:
- Since the environments are taken randomly in the experiments, it is not investigated how sensitive the method is with respect to the choices of environments. Also, does it make any sense to design environments to include related (and not random) classes?
- It seems necessary to include some experiments to assess sensitivity of the interpretation with regard to the small perturbations that are not changing the class label.

**Experience Assessment:**

I have read many papers in this area.

**Review Assessment: Checking Correctness Of Derivations And Theory:**

I assessed the sensibility of the derivations and theory.

**Review Assessment: Checking Correctness Of Experiments:**

I assessed the sensibility of the experiments.

**Review Assessment: Thoroughness In Paper Reading:**

I read the paper at least twice and used my best judgement in assessing the paper.

---

> ### Author Response · Authors · 2019-11-14
> **Reply to reviewer #1**
>
> Thank you for your constructive review. Below we address your concerns.
>
> “- Since the environments are taken randomly in the experiments, it is not investigated how sensitive the method is with respect to the choices of environments. ”
> We partially addressed this concern in the results of table 1 (page 8) that shows the average F1 scores for the classification task over several runs, where each run has a different randomly selected choice of environments. From this table it is clear that the standard deviation in F1 scores is low and in line with the standard deviations of the baselines. This suggests that for sufficiently large $n_e$ and $R$ (the parameters that determine the amount of environments and the maximum amount of attributes per environment) the method is not sensitive with respect to the choices of environments. We also added this sensitivity for different choices of $n_e$ and $R$ to the appendix of the revised version (see appendix A1, tables A1 and A2), where it becomes clear that even for small values, the sensitivity is low.
>
> “Also, does it make any sense to design environments to include related (and not random) classes?”
> The rationale for not designing (handmade) environments is that they require knowledge about what would lead to distinguishing features.  Note that the randomly selected features will lead to many environments that are related to any class, thus ensuring a good choice over any set of features. The idea is inspired by the Random Features approximation as developed in the work of [1] and [2], we have clarified this in the revision (‘Recent work’ in section 2).
>
> “- It seems necessary to include some experiments to assess sensitivity of the interpretation with regard to the small perturbations that are not changing the class label.”
> Our interpretation of your question is as follows, please correct us if necessary: “What amount of perturbation is needed before the interpretation of the class label of the representation changes?”. As part of the retrieval method we evaluated the size of the factor $q$ (the factor that determines how much the representation is modified for the retrieval experiment in section 4.3) and its influence on the change in class. This gives an estimate of the sensitivity as it shows that small perturbations to representations don’t easily modify class membership. We have added the results in section 4.3.
>
> [1] https://arxiv.org/abs/1811.01713
> [2] http://papers.nips.cc/paper/3182-random-features-for-large-scale-kernel-machines.pdf

---

### Comment · Area_Chair1 · 2019-11-14
**Reviewers, any comments on author response?**

Dear Reviewers, thanks for your thoughtful input on this submission!  The authors have now responded to your comments.  Please be sure to go through their replies and revisions.  If you have additional feedback or questions, it would be great to get them this week while the authors still have the opportunity to respond/revise further.  Thanks!

---

### Decision · Program_Chairs · 2019-12-19

**Decision:**

Reject

**Comment:**

The paper proposes an approach for learning class-level and individual-level (token-level) representations based on Wasserstein distances between data subsets.  The idea is appealing and seems to have applicability to multiple tasks.  The reviewers voiced significant concerns with the unclear writing of the paper and with the limited experiments.  The authors have improved the paper, but to my mind it still needs a good amount of work on both of these aspects.  The choice of wording in many places is imprecise.  The tasks are non-standard ones so they don't have existing published numbers to compare against; in such a situation I would expect to see more baselines, such as alternative class/instance representations that would show the benefit specifically of the Wasserstein distance-based approach.  I cannot tell from the paper in its current form whether or when I would want to use the proposed approach.  In short, despite a very interesting initial idea, I believe the paper is too preliminary for publication.